# Correlation Studies between S100 Protein Level and Soluble MIA or Tissue MelanA and gp100 (HMB45) Expression in Cutaneous Melanoma

**DOI:** 10.3390/jpm13060898

**Published:** 2023-05-26

**Authors:** Lucica Madalina Bolovan, Mihai Ceausu, Adina Elena Stanciu, Marieta Elena Panait, Antonela Busca, Camelia Mia Hotnog, Coralia Bleotu, Laurentia Nicoleta Gales, Mihai Teodor Georgescu, Virgiliu Mihail Prunoiu, Lorelei Irina Brasoveanu, Silviu Cristian Voinea

**Affiliations:** 1Carcinogenesis and Molecular Biology Department, Institute of Oncology “Prof. Dr. Alexandru Trestioreanu”, 022328 Bucharest, Romania; madalina.bolovan@iob.ro (L.M.B.); adinaelenastanciu@yahoo.com (A.E.S.); 2Pathology Department, Institute of Oncology “Prof. Dr. Alexandru Trestioreanu”, 022328 Bucharest, Romania; mihail.ceausu@umfcd.ro; 3Cancer Biology Department, Institute of Oncology “Prof. Dr. Alexandru Trestioreanu”, 022328 Bucharest, Romania; me_panait@yahoo.com (M.E.P.); antonelabusca@gmail.com (A.B.); 4Center of Immunology, “Stefan S. Nicolau” Institute of Virology, Romanian Academy, 030304 Bucharest, Romania; camelia.hotnog@virology.ro; 5Cellular and Molecular Pathology Department, “Stefan S. Nicolau” Institute of Virology, Romanian Academy, 030304 Bucharest, Romania; coralia.bleotu@virology.ro; 6Oncology Department, University of Medicine and Pharmacy “Carol Davila” Bucharest, 050474 Bucharest, Romania; laurentia.gales@umfcd.ro (L.N.G.); mihai.georgescu@umfcd.ro (M.T.G.); 7Oncology Department, Institute of Oncology “Prof. Dr. Alexandru Trestioreanu”, 252 Fundeni Ave, 022328 Bucharest, Romania; 8Oncological Surgery Department, University of Medicine and Pharmacy “Carol Davila” Bucharest, 050474 Bucharest, Romania; virgiliu.prunoiu@umfcd.ro (V.M.P.); silviu.voinea@umfcd.ro (S.C.V.); 9Oncological Surgery Department, Institute of Oncology “Prof. Dr. Alexandru Trestioreanu”, 022328 Bucharest, Romania

**Keywords:** cutaneous melanoma, biomarkers, S100B/S100, MIA, MelanA, gp100 (HMB45)

## Abstract

(1) Background: Cutaneous melanoma (CM) originates from melanocytes and causes 90% of skin cancer deaths; therefore, the comparison of different soluble and tissue markers could be valuable in the detection of melanoma progression and therapy monitoring. The present study is focused on the potential correlations between soluble S100B and MIA protein levels in different melanoma stages or with tissue expression of S100, gp100 (HMB45), and MelanA biomarkers. (2) Methods: Soluble S100B and MIA levels were evaluated by means of immunoassay methods in blood samples from 176 patients with CM, while tissue expressions of S100, MelanA, and gp100 (HMB45) were detected by means of immunohistochemistry in 76 melanomas. (3) Results: Soluble S100B correlated with MIA in stages III (*r* = 0.677, *p* < 0.001) and IV (*r* = 0.662, *p* < 0.001) but not in stages I and II; however, 22.22% and 31.98% of stage I and II patients, respectively, had high values for at least one of the two soluble markers. S100 tissue expression correlated with both MelanA (*r* = 0.610, *p* < 0.001) and HMB45 (*r* = 0.476, *p* < 0.01), while HMB45 and MelanA also significantly positively correlated (*r* = 0.623, *p <* 0.001). (4) Conclusions: Blood levels of S100B and MIA corroborated with melanoma tissue markers expression could help to improve the stratification process for patients with a high risk of tumor progression.

## 1. Introduction

Cutaneous melanoma (CM) is one of the most aggressive types of skin cancer once melanoma cells have reached the systemic flow. If CM is diagnosed early and primary tumor tissue is removed, these patients’ relative 5-year survival rate is approximately 90%. For stages I and II, the melanoma-specific survival rate (MSS) is 98% and 90%, respectively, but in stage III, when tumor cells invade the lymph nodes, the survival rate varies from 93% to 32%, depending on the stage III subgroup (IIIA–IIID), and is dramatically reduced to 30% or less when melanoma cells spread to distant parts of the body and become metastatic melanoma in stage IV [1,2,3,4,5]. Therefore, there is a real need to improve this ruthless prognosis for advanced cutaneous melanoma, knowing that early diagnosis in the initial stages is a major factor that significantly increases the chances of disease-free status and a high survival rate [6]. Prospectively, the melanoma markers detected in blood and tumor tissue samples at the time of diagnosis or during the disease follow-up could provide specific tools for the early identification of melanoma patients with aggressive profiles and a high risk of recurrence or metastasis [7,8,9,10,11]. However, in addition to blood testing, tissue evaluation of melanoma markers brings useful information about cytomorphological characteristics that differ among melanoma types and are crucial for diagnosis and disease prognosis, with clinical implications for personalized therapy [12].

One of the most investigated and useful biomarkers in melanoma is S100B, which belongs to the large family of S100 proteins, consisting of numerous isoforms, with more than 25 members sharing structural similarity and functional discrepancy. The S100 family is part of the group of calcium-binding proteins, expressed in cell and tissue-specific manners, and distinguished by structure, conformation, and extracellular and/or intracellular functions from other proteins of this group [13,14,15]. In clinical routines, cellular S100 is used as an immunohistochemical (IHC) tumor marker for melanoma diagnosis, while the soluble form serves as a prognostic biomarker [16,17]. Therefore, the European Society for Medical Oncology (ESMO) recommends the S100B protein as a blood test marker in the follow-up of melanoma patients, pointing out that increased levels of serum S100B protein have both high specificity for disease progression and a significant role in monitoring stage III and IV melanoma patients and in the prognosis of the disease [18]. Different studies performed at the tissue level reported that S100 is one of the most sensitive markers, exhibiting 89–100% positivity for primary melanomas and with a specificity varying from 70% to 79% between melanocytic and non-melanocytic lesions [13,19]. The tissue expression of S100 is strong and diffusely distributed, both in the nucleus and the cytoplasm, with a characteristic pattern [20,21]. To increase the test specificity, it is advisable to perform a simultaneous analysis of S100 with other melanocytic differentiation proteins, such as MelanA (MART-1) melanoma antigen recognized by T cells, and glycoprotein 100 (gp100, pre-melanosome protein 17 (Pmel 17), SILV) [19,22,23].

MelanA (MART-1) is a transmembrane protein detected in the cytoplasm of melanin-producing cells, including normal melanocytes and primary and metastatic melanoma tumors, as well as uveal melanomas [24,25,26]. MelanA has a sensitivity of 83–100%, while the specificity is 81–98% (melanocytic vs. non-melanocytic) in primary melanomas and 71–88% in metastatic lesions [27]. The overall survival and disease-free rates are reduced by the progressive loss of MelanA protein expression from the primary tumor and by the association with a Breslow thickness ≥ 1 mm [25]. The functions of MelanA are related to the gp100 protein by the formation of a complex that influences the structural stability and the maturation of melanosome [28]. MelanA is more specific than the S100 protein and expresses more intense and diffuse staining than gp100 (HMB45) [25,29].

Similarly to S100 and MelanA, gp100 (HMB45) represents a tool to differentiate malignant lesions from benign nevi [30]. The gp100 protein is a 100 kD glycoprotein, a melanocyte-specific type I membrane protein required for passing from stage I to stage II of the maturation of the melanosome structure, and therefore, it is involved in the biogenesis and polymerization of melanin. The gp100 protein is detected by its interaction with the HMB45 mouse anti-human monoclonal antibody, which binds to the glycosylated form of gp100 from pre-melanosome striations [29,30] and is routinely used to visualize the cytoplasmatic marker, mainly in primary cutaneous malignant melanomas [11]. Thus, according to different published data, the sensitivity varies between 72 and 100% in primary melanomas and decreases from 95% to 58% in metastatic lesions, while the specificity is approximately 91–100% for melanocytic vs. non-melanocytic cells [19,25,30].

In addition to the melanoma markers mentioned above, the melanoma inhibitory activity protein (MIA) is another useful clinical biomarker: the protein was purified in 1989, and the coding gene was cloned in 1994. The molecule belongs to a family that comprises four members: MIA, MIA2, TANGO (transport and Golgi organization protein 1), and OTOR (otoraplin), with 34–45% of amino acid structure homology and 47–59% of cDNA sequence homology [31]. MIA is expressed in melanoma cells, and its soluble form is considered a valuable biomarker for the evaluation of melanoma progression [14,32]. MIA is involved in the detachment of melanoma cells from the extracellular matrix (ECM), inhibiting their attachment to fibronectin and laminin and promoting cell migration and invasion. Therefore, the soluble MIA levels correlate with the detachment of cancer cells, local tumor spreading, metastasis, apoptosis inhibition, and disease stabilization/remission or survival prognosis [14,33]. Currently, the role of MIA family members as tumor markers is investigated in melanomas and other neoplastic diseases such as esophageal squamous-cell carcinoma, lung cancer, and cervical cancer [34].

Considering the above aspects, our study aimed to analyze in CM patients the potential correlations between serum S100B and/or tissue S100 protein expression with MIA plasma levels and the tissue expression of gp100 (HMB45) and MelanA (MART-1) markers. Therefore, we set out the following specific objectives: (i) to detect the soluble S100B and MIA protein levels in blood samples collected from melanoma patients in different disease stages; (ii) to find out if soluble S100B correlates with MIA levels, depending on melanoma stages; (iii) to evaluate S100, MelanA (MART-1), and gp100 (HMB45) protein levels of expression in melanoma tissues and analyze the potential correlations between these biomarkers; (iv) to investigate the presence or absence of soluble S100B and/or MIA in patients whose tissues were stained for one, two, or all three tissue biomarkers under study.

## 2. Materials and Methods

### 2.1. Group of Patients

Serum S100B and plasma MIA protein expressions were retrospectively evaluated in a cohort of 176 patients (78F/98M, mean age: 56.2 ± 12.08 years) registered to the “Prof. Dr. Alex. Trestioreanu” Institute of Oncology in Bucharest, Romania and enrolled from 2016 to 2022, who were diagnosed with CM at different stages of the disease, as confirmed by their histopathology analysis. The eligible patients were classified according to the AJCC melanoma staging system [3] at the time of sampling and organized into the following subgroups: stage I (*n* = 27), stage II (*n* = 53), stage III (*n* = 74), and stage IV (*n* = 22). A group of 56 healthy blood donors (25M/31F) aged between 20 and 75 years were used as controls throughout all the immunoassays. Exclusion criteria were applied to select the eligible patients and control subjects. The following groups were excluded: (i) all patients under the age of 18 (*n* = 2); (ii) all patients whose samples were hemolyzed (*n* = 3) or lipemic (*n* = 2); (iii) patients with confirmed rheumatic disease (*n* = 1). The latter criterion helped to discriminate between MIA specific to melanoma cells and other cellular sources, since MIA is physiologically secreted by the chondrocytes, which regulate signaling processes during cartilage differentiation, and its level is significantly increased in rheumatoid arthritis [35].

### 2.2. Blood Sampling and Methods

Blood was drawn from patients after surgery or during the disease follow-up; the samples were processed and stored for further assays. Peripheral blood samples were collected in 5 mL tubes containing a serum clot activator for S100B detection or 3 mL anticoagulant (K_2_EDTA) tubes for plasma separation when MIA protein expression was quantified. Then, samples were centrifuged at 4 °C for 15 min/1000× *g*, and the sera or plasmas were stored at −80 °C until use. The normal blood samples were collected before collecting those from the melanoma patients, and they were stored and processed in the same manner as the patients’ samples described above.

The serum concentration of S100B protein was measured using the CanAg S100 EIA kit (Fujirebio Diagnostics AB, Göteborg, Sweden), a solid-phase two-step non-competitive immunoassay that recognizes S100A1B and S100BB protein epitopes. Plasma MIA protein was quantified using a one-step photometric enzyme-linked immuno-sorbent assay (ELISA) (Roche Diagnostics GmbH, Mannheim, Germany). Both soluble markers were assessed in duplicate in undiluted samples; the standard curves were calculated linearly according to the kit recommendations. The upper limit of normal concentration for plasmatic MIA protein was established using the receiver operating characteristic (ROC) curve at 9.4 ng/mL [36] and 90 ng/L for S100B, as specified by the kit manufacturer.

### 2.3. Tissue Sampling and Staining

Histopathological (HP) assessments were performed on 76 tissue specimens from surgically excised melanomas. The fragments were harvested from different body skin areas, such as the chest, the superior and inferior limbs, the head and neck, and the hips. The selected tissue samples were fixed in 10% neutral buffered formalin (pH 7.0) and paraffin-embedded. Sections were cut at 5 μm and stained with standard hematoxylin and eosin (H&E) for 4–6 h at room temperature (RT). Immunohistochemical analysis (IHC) was performed with a panel of three primary antibodies, using sections displayed on poly-L-lysine treated slides. The panel consisted of the following antibodies: (1) anti-S100 (clone: 4C4.9, ready-to-use (RTU), Ventana, Roche Diagnostics Corporation, Indianapolis, IN, USA), (2) anti-MelanA (clone: A103, RTU, Ventana), and (3) anti-melanosome gp100 (clone: HMB45, RTU, Ventana). IHC was performed on 3 μm thick sections from formalin-fixed paraffin-embedded specimens. The method used was an indirect tristadial avidin-biotin complex (ABC) technique, with a NovoLink polymer detection system that utilized a novel control polymerization technology to prepare polymeric HRP-linker antibody conjugates, according to the manufacturer’s specifications (Novocastra, Leica Biosystems, Deerfield, IL, USA). An antigen retrieval technique was performed for MelanA but not for S100 and HMB45. According to the producer’s specifications, for MelanA, a heat-induced epitope retrieval (HIER) in a Tris-EDTA buffer (pH = 7.8) at 95 °C for 44 min (Ventana) was prepared. All HP and IHC slides were examined and photographed on a Leica DM 2500 microscope (Leica Microsystems, Wetzlar, Germany), and the digital images were acquired using a Leica MC 190 HD incorporated camera (Leica Microsystems). The images were processed and analyzed using the LAS EZ Leica Application Suite v.3.4.0 software program running under Windows 10 (Leica Microsystems).

### 2.4. Statistical Analysis

Statistical analyses of the experimental data were performed using Statistica software (version 8.0; StatSoft, Inc., Tulsa, OK, USA) and GraphPad Prism 7 software (GraphPad Software Inc., La Jolla, CA, USA). The results are presented as mean values ± standard deviation, minimum and maximum values, and percentages. The Pearson *r* correlation coefficient was calculated to discover the potential correlation between the analyzed parameters and the strength of the statistical significance, according to the number of subjects (n). *p*-values < 0.05 were considered statistically significant; *r*-values showed a strong correlation for ±0.5 ≤ *r* ≤ ±1, a moderate correlation for ±0.3 ≤ *r* ≤ ±0.49, and a low correlation for *r* ≤ ±0.29.

## 3. Results

### 3.1. Soluble Levels of S100B and MIA Expression in Melanoma Patients

The immunoassays were performed to detect the serum levels of S100B protein and the plasma levels of MIA protein in blood samples from melanoma patients at different stages of disease progression.

#### 3.1.1. Circulating Levels of S100B and MIA in Melanoma Patients

Experimental data were obtained after performing the immunoassays for detecting the S100B and MIA expression levels in the peripheral blood collected from a study group of 176 melanoma patients. The results were analyzed, and descriptive statistics are presented below in the Table 1. The mean level of serum S100B in the whole melanoma group was 132.04 ± 224.43 ng/L (min. value 15.6 ng/L, max. value 1580.32 ng/L), while the mean level of plasma MIA was 10.88 ± 7.96 ng/mL (min. 4.20 ng/mL, max. 58.61 ng/mL). When the data were analyzed, we noticed that the average concentrations of circulating S100B and MIA were closely comparable in the female and male subgroups (Table 1, Figure 1).

#### 3.1.2. Circulating Levels of S100B and MIA Depending on the Stage of the Disease

Data analyses of the whole group showed that 30.11% and 35.22% of melanoma patients expressed positive values for S100B and MIA, respectively. The levels of S100B serum concentrations were evenly distributed around the cutoff of 90 ng/L: 7.4% of stage I, and 11.3% of stage II patients had higher values than the normal limit value, while the percentages of patients with positive values increased in the advanced stages, reaching 41.8% in stage III and 63.6% in stage IV. Moreover, the mean values of soluble S100B increased with the disease progression, from 67.44 ng/L in stage I to 357 ng/L in stage IV (Table 2). When the data for each melanoma stage were analyzed, we discovered that plasma levels of MIA also gradually increased over the cutoff of 9.4 ng/mL along with the disease progression, with the frequency of the patients who expressed higher MIA being 18.5% for stage I, 22.6% for stage II, 40.7% for stage III, and 59% for stage IV, the increases being greater the more advanced the stage of the disease was. Further correlation analyses between soluble S100B and MIA protein levels showed that both markers had high values in 1.8% of the cases for stages I and II, 16.21% of the stage III cases, and 40.9% of the stage IV cases. It is a significant fact that at least one of the two soluble proteins is overexpressed in melanoma patients, with high levels being found in 22.22%, 31.98%, 66.21%, and 81.81% of the cases in stages I, II, III, and IV, respectively (Table 2, Figure 2).

#### 3.1.3. Correlation Analysis between Circulating S100B and MIA Levels

A statistical analysis of the serological experiments revealed a significant positive strong correlation between circulating S100B and MIA proteins (*r* = 0.672, *p* < 0.001) when the whole group of 176 patients was considered. Moreover, the correlation between S100B and MIA was stronger in the female subgroup (*r* = 0.840, *p* < 0.001) than in the males (*r* = 0.543, *p* < 0.001), as shown in Figure 3.

#### 3.1.4. Correlation between Circulating S100B and MIA Levels in Distinct Stages of the Disease

Statistical analyses showed that the soluble levels of S100B did not correlate with the MIA levels in stages I and II (*r* = −0.108 and *r* = −0.046, *p >* 0.05). In contrast, our results showed a statistically significant positive strong correlation between the S100B and MIA levels in stages III (*r* = 0.677, *p* < 0.001) (Figure 4a) and IV (*r* = 0.662, *p* < 0.001) (Figure 4b).

### 3.2. Tissue Expression of Melanoma Markers

#### 3.2.1. Immunohistochemical Analysis (IHC) of Melanoma Lesions

After the routine histopathological assessment of the tissue specimens from surgically excised melanomas harvested from different body skin areas, IHC staining was performed on 76 melanoma tissues for S100, MelanA, and/or HMB45 detection. The results obtained by the IHC staining assay are presented as follows: (i) individual groups for each marker (S100, MelanA, HMB45); (ii) groups of two associated markers (S100 + HMB45; S100 + MelanA; HMB45 + MelanA); (iii) a group of three tested markers (S100 + HMB45 + MelanA). Each marker group was analyzed depending on age, localization, histological subtype, Breslow thickness, stage, and tissue expression. The results and the clinical characteristics are detailed in Table 3.

An example of IHC analysis performed on a sample excised from a melanoma patient is presented in Figure 5. In this case, a positive staining was observed into the cytoplasm of the tumor cells, for both S100 and MelanA, with a variable focal granular reaction for the latter. HMB45 was also positively stained into the cytoplasm of the tumor cells with a focal granular reaction (Figure 5).

#### 3.2.2. Correlation Analysis between Tissue Expressions of Melanoma Markers

In order to detect the potential correlation between the levels of tissue expression of the S100, MelanA, and gp100 (HMB45) melanoma markers, we performed several statistical data analyses. Therefore, the group of tissues collected from melanoma patients (n = 76) was divided into several subgroups and analyzed according to the number of stained tissue markers on the same samples: (i) three groups of two markers and (ii) a group of three markers.

(i) For each group of two tested tissue markers, the calculated correlation coefficient *r* showed that the S100 tissue protein was strongly correlated with the MelanA tissue marker (*r* = 0.610, *p* < 0.001) and moderately correlated with HMB45 (*r* = 0.476, *p* < 0.01); similarly, the HMB45 protein was also strongly correlated with MelanA (*r* = 0.623, *p* < 0.001) (Table 4).

(ii) The statistical analyses were also applied to data obtained after IHC was performed on 33 melanoma tissues stained for all three histological markers under study (S100, HMB45, and MelanA). On these samples, statistical results showed that S100 tissue expression statistically moderately correlated with HMB45 (*r* = 0.438, *p* < 0.01) and with MelanA (*r* = 0.493, *p* < 0.01), but no significant correlation was found between MelanA and HMB45 expression (*r* = 0.357, *p* > 0.05).

In conclusion, the analyses of the associations between S100 and MelanA, S100 and HMB45, or HMB45 and MelanA from both groups (i and ii) showed that the *r* correlation coefficients and *p*-values from the group with two stained markers were slightly higher than those from the group with three stained markers (Table 4 and Table 5).

#### 3.2.3. Soluble Expression of Melanoma Markers in Different Groups of IHC Tested Melanoma Patients

As shown above (Figure 1, Figure 2, Figure 3 and Figure 4, Table 1 and Table 2), we investigated the S100B and MIA levels in peripheral blood from 176 melanoma patients. Besides the serological assays, tissue samples from 76 patients were also evaluated by means of IHC for tissue S100, HMB45, and/or MelanA staining (Figure 5, Table 3, Table 4 and Table 5). The blood samples used for the detection of the soluble markers were matched with the analyzed tissue samples.

In addition, we analyzed the presence or absence of the soluble markers S100B and/or MIA in the cohort of 76 patients whose tissues were subjected to IHC staining of the three tissue markers under study in order to identify if there might be a potential correlation between the soluble forms of S100B and MIA evaluated in this group (Table 6). When the correlation coefficient was calculated, we found a significant positive strong correlation (*r* = 0.753, *p* < 0.001) between these two soluble proteins that was higher than that in the total group of 176 patients (*r* = 0.672, *p* < 0.001) on whom the serological assays were performed (Table 1, Figure 3).

## 4. Discussion

Cutaneous melanoma remains one of the most aggressive forms of cancer with an immunogenic character, a high recurrence rate, a high risk of metastasis, and resistance to therapy [37]. It is known that 90% of all melanoma metastases occur in the first 5 years after surgery; therefore, the follow-up examination is essential at that time, and it is necessary to establish a specific clinical follow-up program depending on the characteristics of the tumor cells from the primary site [38]. Adding the blood and/or tissue tumor biomarkers investigation to the disease evaluation contributes to an increase in the accuracy of the diagnosis, early knowledge of the prognosis factors for the risk of recurrence or metastases, and the selection of optimal therapeutic choices for melanoma patients. However, no melanoma marker fulfills all the attributes of an ideal marker [39].

Many studies have stated that S100 proteins are predominantly upregulated and specific to the histological type, or even subtype, and the disease stage [16,17,40,41]. Therefore, the associations of S100B with two or more specific markers, such as MIA, HMB45, and MelanA, and/or any other clinical changes, such as lymph node status [42,43,44], might offer objective and reliable information about disease progression. The involvement of S100 proteins in cell growth, cell cycle regulation, and the differentiation process, with, on the other hand, the role played by MIA proteins in cell detachment, migration, and invasion, or the suppressing of cancer cell apoptosis, justifies and explains the importance of detecting any imbalance in the expression of the two proteins, increasing the chance of choosing the optimal time for investigations and therapeutic approaches [5,24]. Therefore, we analyzed in CM the interrelationship between the S100B and MIA blood levels and also between the tissue expression of the S100 protein and the other two specific melanoma markers, MelanA and gp100 (HMB45). Finally, we also investigated the relationship between the S100B and MIA soluble markers and the tissue melanoma markers under study (S100, MelanA and gp100 (HMB45)), emphasizing the value of these biomolecules as suitable candidates to increase the efficiency of disease evaluation.

In melanoma there are twelve S100 protein members expressed, seven of which are upregulated, a feature that is closely related to tumor growth, angiogenesis, metastasis, and drug resistance [16,45,46]. At the cellular level, the S100B protein is present in the cytoplasm in various cells: astrocytes, melanocytes, adipocytes, chondrocytes, lymphocyte populations, Schwann cells, and oligodendrocytes [47], and, due to multiple and diverse interactions with many types of protein targets at extracellular or intracellular level, it is involved in calcium homeostasis, transcription, the regulation of cell cycles and cell growth, the stimulation of cell proliferation, migration, the inhibition of apoptosis, and differentiation [19,48,49].

At the systemic level, the S100B protein, along with lactate dehydrogenase (LDH), is routinely investigated in melanoma management. However, in recent years, a more sensitive melanoma marker, MIA (melanoma inhibitory activity), has been considered for disease management in patients with metastatic disease or with a profile that is unresponsive to chemotherapy [21]. On the other hand, according to Hofmann M.A. et al., MIA is a prognostic factor in melanoma monitoring, and it seems to be the best predictive biomolecule for tumor migration through the sentinel lymph node, correlated with the number of affected nodes [44].

Based on all these scientific data, circulating MIA protein levels could bring more consistency to the clinical investigation panel. Thus, our results showed that by performing simultaneous measurements of S100B and MIA concentrations in blood, we obtained a significant positive strong correlation between the two parameters in stages III (*r* = 0.677, *p* < 0.001, Figure 4a) and IV (*r* = 0.662, *p* < 0.001, Figure 4b) of melanoma patients, while in stages I and II no significant correlation was observed. In addition, a high percentage of stage III and IV patients have at least one of the two markers overexpressed (66.21%), with higher values being found in patients with metastatic lesions (81.81% of stage III cases and 78.68% of stage IV cases). Lagares A.D. et al. published similar data evaluating the potential clinical use of MIA and S100B as tumor markers in the advanced stages, and it was shown that 75.7% of patients with distant metastases and 63.2% in stage IIIC had higher levels for either S100B or MIA soluble proteins [50,51]. However, we found that in stages I and II, the two biomolecules do not correlate, but the percentage of samples with values above the normal limit suggests that any higher value might indicate a potential risk of unfavorable evolution, and it should be of interest for disease monitoring.

Because CM is an aggressive and unpredictable neoplasm, any measurable molecular changes help to detect the early tumor profile in stages I or II [9,38]. For example, several studies emphasize the role of different molecules such as vascular endothelial growth factor proteins (VEGF), matrix metalloproteinases (MMP-2 and 9), and tumor-related tissue inhibitory metalloproteinases (TIMPs) that express higher serum values in patients with early-stage melanoma [52,53]. Analyzing the gender subgroups, we noticed that S100B and MIA soluble proteins are better correlated in the female group than in the male group. Currently, this finding needs more data to justify and prove the potential link between these proteins and gender specificity.

Our investigations performed on melanoma tissues showed that in the group of 33 patients in which all three tissue markers (S100, MelanA, and HMG45) were tested, S100 was the most sensitive marker for melanoma, being positive in 96.96% of the tissues (32/33). Additionally, 78.78% of the cases (26/33) presented positive staining for all three tissue markers. Moreover, 90.9% of the patients after tumor excision had a serum S100B level below the upper limit; these data support the sensitivity and specificity of the S100B protein in melanomas [20]. Among the cases with a triple positive expression of tissue proteins, 11.53% (3/26) continued to overexpress serum S100B after surgery, reflecting the possibility of the occult presence of melanoma cells as a source of S100B protein. These patients are considered at significant risk of relapse or tumor cell invasion and should be closely monitored regarding their body skin, lymph nodes, and mucous membranes [54]. On the other hand, the histological characteristics and the expression of melanoma-specific proteins provide crucial data on the prognosis of the disease. Thus, Berset M. et al. showed that if MelanA immunostaining is positive and Breslow thickness ≥ 1 mm, 74% of patients have an overall survival rate of approximately 5 years, but if MelanA is negative, the 5-year survival rate of patients is only 17%. If the Breslow value is ≤1 mm, then the prognosis increases to 96% of patients for 5 to 10 years, regardless of MelanA expression [28]. Consequently, the decrease or loss of MelanA expression in the transition to metastatic status indicates a poor prognosis [12]. From our results, it appears that when comparing the data between the two groups with two and three stained markers, we discovered that the associations between S100/MelanA, S100/HMB45, and HMB45/MelanA show a stronger correlation if analyzed in the group with two stained markers rather than in the group with all three tested markers. This finding might suggest that one of the three proteins reduces the statistical significance, but more data analyses are needed to demonstrate this.

## 5. Conclusions

The expression of melanoma tissue S100, MelanA, and HMG45 biomarkers, corroborated with the blood levels of S100B, and MIA, could be reliable factors in the stratification of patients with a high risk of progression. The limitation of this study is its retrospective feature, but the perspective results could help to implement and increase the efficacy of melanoma management. More data, and further appropriate statistical analyses performed on increased groups of melanoma patients, might provide additional accurate information on the impact of blood and/or tissue biomarkers on therapeutic approaches, increasing the disease-free and survival rates of cutaneous melanoma patients.

## Figures and Tables

**Figure 1 jpm-13-00898-f001:**
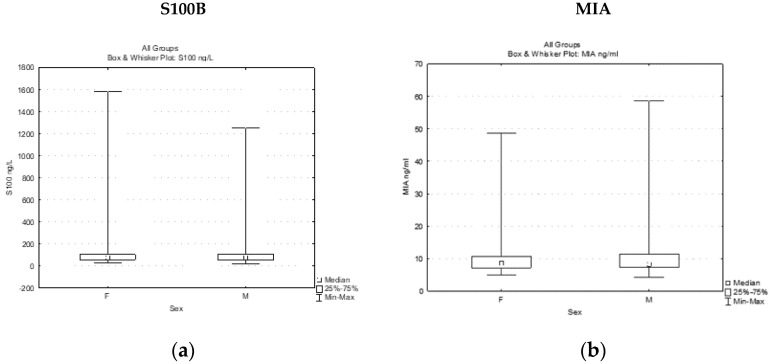
Soluble S100B (**a**) and MIA (**b**) levels in female and male melanoma patients.

**Figure 2 jpm-13-00898-f002:**
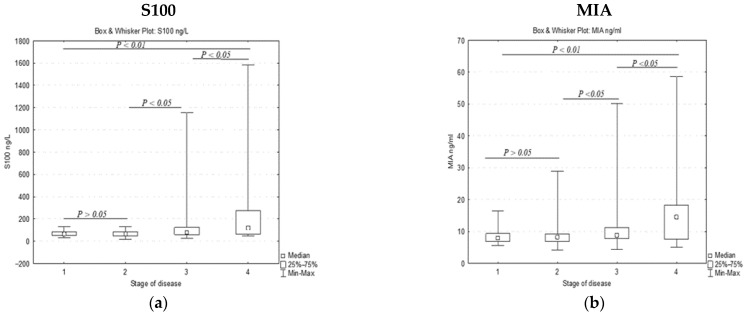
Soluble S100B (**a**) and MIA (**b**) levels in patients in different melanoma stages.

**Figure 3 jpm-13-00898-f003:**
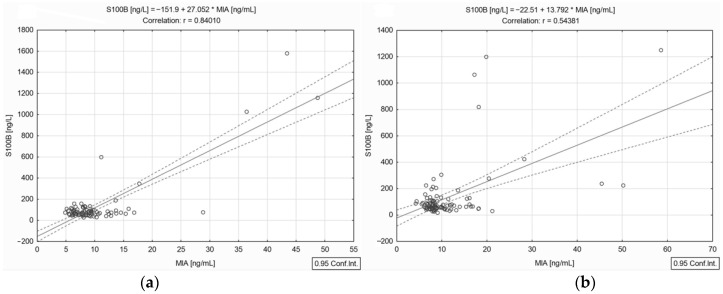
Correlation between circulating S100B and MIA levels in female (**a**) and male subgroups (**b**) (— fitted linear regression curve; --- equality line).

**Figure 4 jpm-13-00898-f004:**
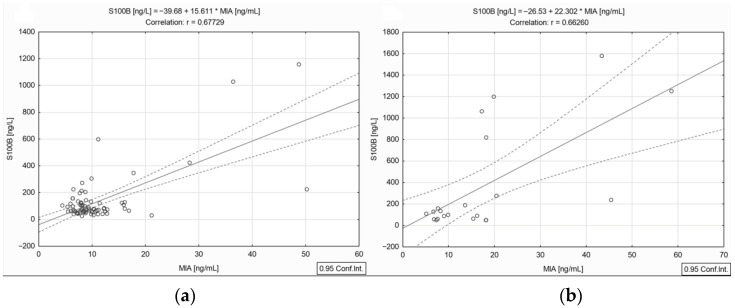
Correlation between circulating S100B and MIA levels in stage III (**a**) and IV (**b**) melanoma patients (— fitted linear regression curve; --- equality line).

**Figure 5 jpm-13-00898-f005:**
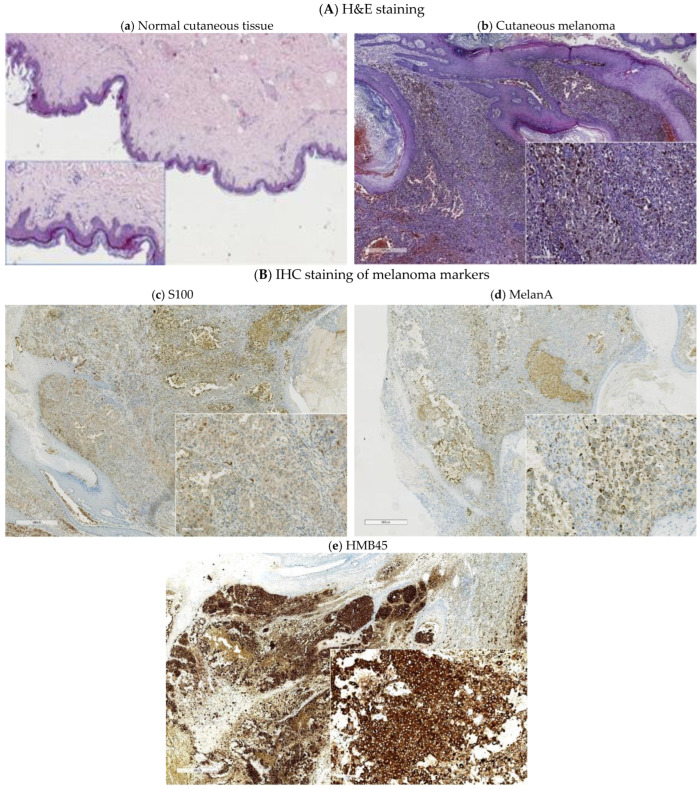
Images of histopathological and IHC analyses: (**A**) Normal cutaneous tissue (**a**) vs. superficial spreading malignant melanoma (**b**) (H&E staining, overview with a low power field (40×), inset: detailed image (200×)); (**B**) expression of tissue markers analyzed by IHC in a sample excised from a melanoma patient; (**c**) S100 staining in tumor cell (40×), inset: detailed image with weak reaction in cytoplasm (200×); (**d**) MelanA focal staining (40×), inset: detailed image with cytoplasmic granular expression (200×); (**e**) HMB45 staining (40×), inset: detailed image with strong cytoplasmic reaction (200×).

**Table 1 jpm-13-00898-t001:** Circulating S100B and MIA levels in female and male melanoma subgroups and entire melanoma group.

Sex	Female	Male	Melanoma Group
Number of patients	78	98	176
S100B ng/L (mean ± SD)	132.92 ± 241.86	131.33 ± 210.81	132.04 ± 224.43
Minimum	25.53	15.61	15.61
Maximum	1580.32	1250	1580.32
MIA ng/mL (mean ± SD)	10.61 ± 7.46	11.09 ± 8.37	10.88 ± 7.968
Minimum	4.89	4.20	4.20
Maximum	48.74	58.61	58.61

**Table 2 jpm-13-00898-t002:** Statistical characteristics for serum S100B and plasma MIA proteins depending on the melanoma stages.

	No. of Patients	Mean ± SD	Minimum	Maximum
S100B (cut-off 90 ng/L)				
Stage I	27	67.44 ± 21.71	31.33	128.60
Stage II	53	66.43 ± 23.20	15.61	132.30
Stage III	74	135.71 ± 186.10	25.53	1157.00
Stage IV	22	357.00 ± 477.65	47.55	1580.32
MIA (cut-off 9.4 ng/mL)				
Stage I	27	8.61 ± 2.55	5.59	16.49
Stage II	53	8.92 ± 3.73	4.20	28.85
Stage III	74	11.23 ± 8.07	4.42	50.21
Stage IV	22	17.19 ± 14.19	5.11	58.61

**Table 3 jpm-13-00898-t003:** IHC expression of S100, HMB45, and/or MelanA tissue markers in various groups of melanoma patients classified by clinico-pathological characteristics.

No	Clinico-Pathogical Characteristics	S100^+^ *n* (%)	HMB45^+^ *n* (%)	MelanA^+^ *n* (%)	S100^+^/HMB45^+^ *n* (%)	S100^+^/MelanA^+^ *n* (%)	HMB45^+^/MelanA^+^ *n* (%)	S100^+^/HMB45^+^/ MelanA^+^ *n* (%)
1	Number of patients (*n*)	51	61	65	37	42	55	33
2	Age							
	<65 years	32 (62.75%)	34 (55.74%)	39 (60%)	21 (56.75%)	25 (59.53%)	32 (58.18%)	19 (57.57%)
	>65 years	19 (37.25%)	27 (44.26%)	26 (40%)	16 (43.25%)	17 (40.47%)	23 (41.82%)	14 (42.43%)
3	Sex							
	Males	24 (47.05%)	33 (54.09%)	33 (50.77%)	20 (54.05%)	20 (47.62%)	29 (52.73%)	16 (48.48%)
	Females	27 (52.95%)	28 (45.91%)	32 (49.23%)	17 (45.95%)	22 (52.38%)	26 (47.27%)	17 (51.52%)
4	Localization							
	Chest	29 (56.87%)	32 (52.45%)	32 (49.23%)	22 (59.46%)	23 (54.77%)	30 (54.54%)	21 (63.64%)
	Inferior limbs	12 (23.52%)	13 (21.31%)	16 (24.61%)	7 (18.92%)	10 (23.81%)	10 (18.18%)	5 (15.15%)
	Superior limbs	7 (13.72%)	9 (14.75%)	10 (15.38%)	5 (13.51%)	6 (14.28%)	8 (14.55%)	4 (12.12%)
	Head and neck	2 (3.93%)	4 (6.55%)	4 (6.16%)	2 (5.40%)	2 (4.76%)	4 (7.28%)	2 (6.06%)
	Hips	1 (1.96%)	3 (4.91%)	3 (4.62%)	1 (2.71%)	1 (2.38%)	3 (5.45%)	1 (3.03%)
5	Histological subtype							
	Superficial spreading	38 (74.50%)	45 (73.77%)	48 (73.84%)	25 (67.56%)	29 (69.04%)	39 (70.90%)	16 (48.49%)
	Nodular	12 (23.53%)	14 (22.95%)	14 (21.54%)	12 (32.44%)	12 (28.57%)	14 (25.46%)	14 (42.42%)
	Acral	1 (1.97%)	1 (1.64%)	2 (3.08%)	-	1 (2.39%)	1 (1.82%)	2 (6.06%)
	Lentigo	-	1 (1.64%)	1 (1.54%)	-	-	1 (1.82%)	1 (3.03%)
6	Breslow thickness							
	<1 mm	10 (19.61%)	8 (13.11%)	7 (10.77%)	4 (10.81%)	4 (9.52%)	5 (9.09%)	2 (6.06%)
	1–2 mm	8 (15.69%)	17 (27.87%)	18 (27.69%)	7 (18.91%)	11 (26.19%)	14 (25.46%)	4 (12.12%)
	2–4 mm	15 (29.41%)	23 (37.70%)	26 (40%)	18 (48.65%)	15 (35.71%)	24 (43.63%)	17 (51.52%)
	>4 mm	18 (35.29%)	13 (21.32%)	14 (21.44%)	8 (21.63%)	12 (28.58%)	12 (21.8%)	10 (30.30%)
7	Stage							
	I	12 (23.53%)	14 (22.95%)	15 (23.08%)	8 (21.62%)	10 (23.80%)	12 (21.82%)	7 (21.21%)
	II	26 (50.98%)	29 (47.54%)	30 (46.15%)	19 (51.35%)	20 (47.62%)	27 (49.09%)	17 (51.52%)
	III	12 (23.53%)	16 (26.23%)	18 (27.69%)	9 (24.32%)	11 (26.19%)	15 (27.27%)	9 (27.27%)
	IV	1(1.96%)	2 (3.28%)	2 (3.08%)	1 (2.71%)	1 (2.39%)	1 (1.82%)	-
8	Tissue expression							
	Positive	48 (94.12%)	52 (85.25%)	56 (86.15%)	-	-	-	-
	Negative	3 (5.88%)	9 (14.75%)	9 (13.85%)	-	-	-	-
	+/+	-	-	-	32 (86.48%)	37 (88.09%)	45 (81.82%)	-
	−/−	-	-	-	1 (2.70%)	2 (4.76%)	5 (9.09%)	-
	+/−	-	-	-	4 (10.82%)	2 (4.76%)	2 (3.64%)	-
	−/+	-	-	-	-	1 (2.39%)	3 (5.45%)	-
	+/+/+	-	-	-	-	-	-	26 (78.79%)
	−/−/−	-	-	-	-	-	-	1 (3.03%)
	+/−/+, +/−/−, +/+/−	-	-	-	-	-	-	6 (18.18%)

MIA: melanoma inhibitory activity; S100: S100 tissue protein; S100B: S100B soluble protein; HMB45: human melanoma black-45 antibody that recognizes gp100 protein; MelanA (MART-1): melanoma antigen recognized by T cells.

**Table 4 jpm-13-00898-t004:** Correlation analyses between expressions of tissues markers S100, HMB45, and MelanA in each group with two tested markers.

	S100 vs. HMB45	S100 vs. MelanA	HMB45 vs. MelanA
Number of patients	37	42	55
Pearson’s *r* correlation coefficient	0.476	0.610	0.623
*p*-value	<0.01	<0.001	<0.001

S100: S100 tissue protein; HMB45: human melanoma black-45 antibody that recognizes gp100 protein; MelanA (MART-1): melanoma antigen recognized by T cells.

**Table 5 jpm-13-00898-t005:** Correlation analyses between expressions of tissue markers S100, HMB45, and MelanA in the group of 33 patients with all three tested biomarkers.

	S100 vs. HMB45	S100 vs. MelanA	HMB45 vs. MelanA
Number of patients (*n* = 33)			
Pearson’s *r* correlation coefficient	0.438	0.493	0.357
*p*-value	<0.01	<0.01	>0.05

S100: S100 tissue protein; HMB45: human melanoma black-45 antibody that recognizes gp100 protein; MelanA (MART-1): melanoma antigen recognized by T cells.

**Table 6 jpm-13-00898-t006:** Soluble S100B and MIA expression in groups of melanoma patients divided by IHC staining of S100, HMB45, and/or MelanA tissue markers.

Tissue Melanoma Markers	IHC Cohort *n* (%)	S100 *n* (%)	HMB45 *n* (%)	MelanA *n* (%)	S100+ HMB45 *n* (%)	S100+ MelanA *n* (%)	HMB45+ MelanA *n* (%)	S100 + HMB45+ MelanA *n* (%)
Number of patients	76	51	61	65	37	42	55	33
Soluble MIA								
Positive	31 (40.79%)	22 (43.14%)	24 (39.34%)	28 (43.07%)	15 (40.54%)	19 (45.24%)	23 (41.82%)	14 (42.42%)
Negative	45 (59.21%)	29 (56.86%)	37 (60.66%)	37 (58.46%)	22 (59.46%)	23 (54.76%)	32 (58.18%)	19 (57.58%)
Soluble S100B								
Positive	15 (19.74%)	11 (21.57%)	7 (11.48%)	12 (18.46%)	3 (8.11%)	8 (19.05%)	7 (12.73%)	3 (9.10%)
Negative	61 (80.26%)	40 (78.43%)	54 (88.52%)	53 (81.53%)	34 (91.89%)	34 (80.95%)	48 (82.27%)	30 (90.90%)
Soluble S100B + MIA								
S100B+/MIA+	9 (11.84%)	7 (13.73%)	4 (6.56%)	8 (12.31%)	2 (5.41%)	6 (14.29%)	4 (7.27%)	2 (6.06%)
S100B+/MIA−	6 (7.89%)	4 (7.84%)	3 (4.92%)	4 (6.15%)	1 (2.70%)	2 (4.76%)	3 (5.45%)	1 (3.03%)
S100B−/MIA+	22 (28.95%)	15 (29.41%)	20 (32.79%)	20 (30.77%)	13 (35.14%)	13 (30.95%)	19 (34.55%)	12 (36.36%)
S100B−/MIA−	39 (51.32%)	25 (49.02%)	34 (55.73%)	33 (50.77%)	21 (56.75%)	21 (50%)	29 (52.73%)	18 (54.55%)

## Data Availability

Data are contained within the article.

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
