# Peer review of "Correlation Studies between S100 Protein Level and Soluble MIA or Tissue MelanA and gp100 (HMB45) Expression in Cutaneous Melanoma"

_jpm, 2023, doi:10.3390/jpm13060898_

Round 1
Reviewer 1 Report
Lucica et al., employ immunoassays to measure the blood levels of S100B and MIA, and tissue levels of S-100, MelanA, and gp100 (HMB-45) proteins in patient samples obtained from participants suffering from different melanoma stages. The authors report a significant correlation between some of the blood and tissue markers of melanoma, suggesting that their evaluation could offer an opportunity to improve the stratification of patients with melanoma.
The paper is recommended for publication, subject to addressing the following comments/concerns.
1. Expression analysis of soluble proteins in this study has been conducted across different melanoma stages. The authors identify markers in the blood sample to be expressed at different levels in different melanoma stages, which provides very valuable information about the utility of using those blood markers as predictors of melanoma progression or stage. Why a similar approach has not been followed in the tissue samples? Separating the tissue staining of proteins at different melanoma stages will add more value to this paper, and it will help to directly compare the blood and tissue markers at different melanoma stages.
2. Samples: Please specify if the blood samples and tissue samples are matched – i.e., coming from the same patients. Also, it is worthwhile to state how the normal tissues (both blood and tissues, if any) were collected. Were they collected around the same time as the collection of melanoma samples? If not, how it has been made sure that the controls are comparable with the tumour samples in terms of storage, processing, etc?
3. It is not clear why specific protein markers were measured in the blood samples and different markers were measured in the tissues. What was the rationale behind this selection? Could the same markers have been evaluated in both sample types?
4. Line 193: It needs to be specified which antibodies were subjected to antigen retrieval and which ones were not. The antigen retrieval conditions need to be provided.
5. Table 2: The normal tissue staining is not provided. The staining of normal tissue samples would help to see where the staining of tumour stages stands in comparison to the control groups.
6. Figure 5: Looking at the images, to me, it seems that the staining has been performed under different conditions, or the images are processed differently, which makes the comparison between the lesions difficult. Since they are all melanoma lesions, the H&E counterstain, for example between c and d, could be similar. Also, the images do not show how the staining looks across the whole tissue sections. I would suggest using lower-magnification images of the tissues to give a better idea about the overall staining, and within the same image, provide a smaller magnified image so that the level and localisation of staining can be seen better.
7. Also, I believe this manuscript requires proofreading. There are some grammatical errors, and some sentences require re-phrasing.
This manuscript requires proofreading. There are some grammatical errors, and some sentences require re-phrasing.
Author Response
“Suggestions for Authors
Lucica et al., employ immunoassays to measure the blood levels of S100B and MIA, and tissue levels of S-100, MelanA, and gp100 (HMB-45) proteins in patient samples obtained from participants suffering from different melanoma stages. The authors report a significant correlation between some of the blood and tissue markers of melanoma, suggesting that their evaluation could offer an opportunity to improve the stratification of patients with melanoma.
The paper is recommended for publication, subject to addressing the following comments/concerns.”
We highly appreciated your critical comments and suggestions. Please find below our responses:
- Expression analysis of soluble proteins in this study has been conducted across different melanoma stages. The authors identify markers in the blood sample to be expressed at different levels in different melanoma stages, which provides very valuable information about the utility of using those blood markers as predictors of melanoma progression or stage. Why a similar approach has not been followed in the tissue samples? Separating the tissue staining of proteins at different melanoma stages will add more value to this paper, and it will help to directly compare the blood and tissue markers at different melanoma stages.
Response:
We identified in the blood samples obtained from melanoma patients, various levels of soluble markers expressed at different melanoma stages; these data bring valuable information about their utility as predictors of melanoma progression or staging. A similar approach has been followed in presenting the data related to tissue staining, as shown in the Table 3, where the 7th group of data show the percentage values of expression for melanoma tissue markers under study at different melanoma stages (Stage I, II, III and IV).
- Samples:Please specify if the blood samples and tissue samples are matched – i.e., coming from the same patients. Also, it is worthwhile to state how the normal tissues (both blood and tissues, if any) were collected. Were they collected around the same time as the collection of melanoma samples? If not, how it has been made sure that the controls are comparable with the tumour samples in terms of storage, processing, etc?
Response:
- The blood samples used for soluble markers detection were matched with the analyzed tissue samples, the blood being collected from the same patient. Therefore, we have added this comment both in the section 2. Materials and Methods, and section 3. Results, 3.2. Tissue expression of melanoma markers, 3.2.3. Soluble expression of melanoma markers in different groups of IHC tested melanoma patients.
- The normal blood samples were collected before collecting the samples from melanoma patients, but they were stored and processed in the same manner as the patients’ samples. The comment was inserted in the section 2. Materials and Methods, 2.2. Blood sampling and methods.
- The apparent normal tissues (tumor margins) from each melanoma tissue samples were not investigated in the present study, but an image with H&E staining of a normal cutaneous tissue was inserted in Figure 5.
- It is not clear why specific protein markers were measured in the blood samples and different markers were measured in the tissues. What was the rationale behind this selection? Could the same markers have been evaluated in both sample types?
Response:
As we have mentioned in section 1. Introduction, the tissue markers studied by us are known as melanocytic markers of differentiation, used in the diagnosis of melanoma, with significant specificity. The S100 protein was analysed both in the blood and tissue samples, while MIA protein was detected only in blood and it was not stained in tissues, since it has not been included yet in the clinical practice according to the guidelines for melanoma management, like the three tissue molecules analyzed in the current study (S100, HMB45 and MelanA). The aim of our study was also to show the benefit of using minimally invasive tests in the diagnosis and prognosis of cutaneous melanoma, such as detection of soluble markers, a fact also due to the accessibility of blood samples from melanoma patients, and the reduced discomfort for them. Previous scientific published data regarding the involvement of MIA in melanoma progression prompted us to consider the soluble MIA protein as a biomolecule that could improve the clinical evaluation along with S100.
- Line 193:It needs to be specified which antibodies were subjected to antigen retrieval and which ones were not. The antigen retrieval conditions need to be provided.
Response:
Melan-A was subjected to antigen retrieval, but not S-100 and HMB-45. According to manufacturer specifications, for Melan-A it was performed a heat induced epitope retrieval (HIER) in Tris-EDTA buffer (pH = 7.8) at 95oC for 44 min (Ventana/ Roche Diagnostics Corporation, Indianapolis, USA). The comment was inserted in section 2. Materials and Methods.
- Table 2:The normal tissue staining is not provided. The staining of normal tissue samples would help to see where the staining of tumour stages stands in comparison to the control groups.
Response:
An image of normal tissue staining was inserted in Figure 5, Panel A, and details given in the figure legend.
- Figure 5:Looking at the images, to me, it seems that the staining has been performed under different conditions, or the images are processed differently, which makes the comparison between the lesions difficult. Since they are all melanoma lesions, the H&E counterstain, for example between c and d, could be similar. Also, the images do not show how the staining looks across the whole tissue sections. I would suggest using lower-magnification images of the tissues to give a better idea about the overall staining, and within the same image, provide a smaller magnified image so that the level and localisation of staining can be seen better.
Response:
We have inserted images for melanoma H&E staining and IHC analyses for the three tissue markers under study, using both a lower-magnification (40x) to give a better idea about the overall staining, and within the same image, we provided a smaller magnified image(200x) so that the level and localisation of staining could be seen better.
- Also, I believe this manuscript requires proofreading. There are some grammatical errors, and some sentences require re-phrasing.
Response:
Our manuscript has been reviewed by a professional translator: several grammatical errors were corrected, and some sentences were re-phrased.

Reviewer 2 Report
In this study, correlation studies were performed between S100 protein level and expression of soluble MIA or tissue MelanA and gp100 (HMB45) in cutaneous melanoma.
Since the article deals with the correlation of serum markers and tissue immunohistochemical markers, it may be helpful for clinicians in terms of melanoma progression and treatment follow-up.
1.The general information in the introduction section can be shortened.
2. Pearson correlation test results are interpreted as follows: High degree: If the coefficient value lies between ± 0.50 and ± 1, then it is said to be a strong correlation. Moderate degree: If the value lies between ± 0.30 and ± 0.49, then it is said to be a medium correlation. Low degree: When the value lies below + . 29, then it is said to be a small correlation.
In the discussion section, it should be emphasized whether there is a moderate or strong correlation.
3. General information about the s100 protein should be abbreviated in the discussion section.
Author Response
Reviewer 2:
“In this study, correlation studies were performed between S100 protein level and expression of soluble MIA or tissue MelanA and gp100 (HMB45) in cutaneous melanoma. Since the article deals with the correlation of serum markers and tissue immunohistochemical markers, it may be helpful for clinicians in terms of melanoma progression and treatment follow-up.”
We highly appreciated your critical comments and suggestions. Please find below our responses:
1.The general information in the introduction section can be shortened.
Response:
We have shortened the general information in the introduction section
- Pearson correlation test results are interpreted as follows: High degree: If the coefficient value lies between ± 0.50 and ± 1, then it is said to be a strong correlation. Moderate degree: If the value lies between ± 0.30 and ± 0.49, then it is said to be a medium correlation. Low degree: When the value lies below + 0.29, then it is said to be a small correlation.
In the discussion section, it should be emphasized whether there is a moderate or strong correlation.
Response:
We have interpretated the Pearson correlation results, as strong or moderate, depending on the values of the coefficient of correlation r, to emphasize the levels of correlation between the obtained experimental data. Comments were included in the Results section, 3.1.4. Correlation between circulating S100B and MIA levels in distinct stages of the disease, 3.2.2. Correlation analysis between tissue expressions of melanoma markers, and 3.2.3. Soluble expression of melanoma markers in different groups of IHC tested melanoma patients.
- General information about the s100 protein should be abbreviated in the discussion section.
Response:
General information about the S100 protein were abbreviated in the discussion section.

Round 2
Reviewer 1 Report
The authors have sufficiently addressed my comments.